# Dengue Severity Prediction in a Hyperendemic Region in Colombia

**DOI:** 10.3390/v17060740

**Published:** 2025-05-22

**Authors:** Jorge Emilio Salazar Flórez, Katerine Marín Velasquez, Luz Stella Giraldo Cardona, Ángela María Segura Cardona, Berta Nelly Restrepo Jaramillo, Margarita Arboleda

**Affiliations:** 1Epidemiology and Biostatistics Group, CES University, Medellín 050001, Colombia; asegura@ces.edu.co; 2Medicine Program, Group for the Research of Infectious and Chronic Diseases (GEINCRO), School of Health Sciences, San Martín University Foundation, Calle 75 sur #34–50, Bloque 2, Sabaneta 055450, Colombia; luz.giraldo@sanmartin.edu.co; 3Colombian Institute of Tropical Medicine, CES University, Medellín 050001, Colombia; kmarin@ces.edu.co (K.M.V.); brestrepo@ces.edu.co (B.N.R.J.); marboleda@ces.edu.co (M.A.)

**Keywords:** severe dengue, prediction model, dengue, warning signs, risk factors

## Abstract

Background: Early detection of severe dengue (SD) is crucial in preventing life-threatening complications. Despite its importance, comprehensive knowledge about these early indicators is still limited. This study aimed to identify predictors of SD in a hyperendemic region of Colombia. Methods: A cross-sectional analysis was conducted using data from 2018 to 2022, encompassing 233 patients. By utilizing the 2009 World Health Organization dengue classifications, cases were differentiated between severe dengue (SD) and non-severe dengue (non-SD). Among these, 47 were confirmed as SD. Associations between clinical, demographic, and laboratory data and disease severity were examined using Fisher’s exact tests or the Mann–Whitney U test (*p* < 0.05). Profiles for SD and non-SD cases were established through multiple correspondence analysis, and a logistic regression-based predictive model was validated using training and test sets. The model’s performance was evaluated using the area under the receiver operating characteristic curve (AUC-ROC), accuracy, sensitivity, F1-score, and precision. Results: Differences in place of residence, comorbidities, type of infection, and signs and symptoms were observed between the severe dengue (SD) and non-severe dengue (non-SD) groups. Median levels of platelets, white blood cells (WBC), aspartate aminotransferase (AST), and alanine aminotransferase (ALT) were found to be higher in the SD group compared to the non-SD group. Neutrophils, leukocytes, platelets, AST, and primary infection were significant predictors of SD. The model demonstrated an area under the receiver operating characteristic curve (AUC) of 0.91 (95% CI, 0.85–0.96). Conclusions: The developed predictive model provides significant assistance to clinicians in assessing SD risk and optimizing triage, which is particularly crucial during dengue outbreaks.

## 1. Introduction

Dengue fever, transmitted by female Aedes mosquitoes, is a systemic infectious disease that is now endemic in over 125 countries [1]. The dengue virus manifests in four distinct serotypes (*DENV-1*, *DENV-2*, *DENV-3*, and *DENV-4*), each varying in terms of virulence, severity, and epidemic potential [2,3]. Annually, between 5 and 10% of acute dengue fever cases progress to severe dengue (SD), a serious complication characterized by plasma leakage, hemorrhage, shock, and potential organ damage [4,5]. Notably, the mortality rate of dengue shock syndrome (DSS) is estimated to be 50 times higher than that of dengue fever alone [6].

The World Health Organization’s (WHO) 2009 classification system divides dengue into three categories: dengue with warning signs (DWS), dengue without warning signs (DWOS), and severe dengue (SD) [5]. Severe symptoms typically emerge between days 4 and 6 of the illness, marked by hypotension, shock, and potential hemorrhaging [5,7]. Hospitalization is advised for individuals exhibiting symptoms of DWS, as these can appear late and hinder early detection. Additionally, hospitalization should be prioritized for patients who may not show warning signs but are in special conditions, such as pregnancy or having comorbidities, as well as for patients categorized as group B2 who are at risk of entering the critical phase [5]. The patients in group A may be cared for at home because they do not exhibit any warning signs. The patients in group B have warning signs or coexisting conditions, such as pregnancy, infancy, old age, diabetes mellitus, renal failure, or social circumstances such as living alone or living far from the hospital [5]. With no specific treatments for dengue, the early detection of disease progression is crucial [8,9]. Following a DENV infection diagnosis, accurately predicting the progression to SD is essential for effective resource allocation and disease management.

Globally, an estimated 390 million dengue infections are reported each year, resulting in over 90 million acute cases and approximately 0.5 million deaths [10]. In Latin America, dengue is becoming an increasingly critical global public health issue, particularly due to rising temperatures and increased rainfall, which create ideal breeding conditions for mosquitoes [11]. From 2014 to 2022, South American countries accounted for 84% of the 17 million cases reported to the Pan American Health Organization (PAHO), with Brazil and Colombia contributing 32% and 29% of the 4203 severe dengue cases, respectively [12]. In 2019, Colombia experienced an epidemic with an incidence rate of 465.9 per 100.000 population, surpassing the 2016 rate of 313.5 per 100,000 population [13]. The subsequent year saw Colombia recording 76,220 dengue cases, with 862 classified as severe. In 2024, 0.93% of dengue cases (*n* = 320,982) were classified as severe dengue, and among these, 95.1% required hospitalization [14].

The research participants were from Apartadó, Turbo, and Chigorodó, three of the eleven towns in the Urabá region, situated within the department of Antioquia, Colombia. In 2022, the Antioquia region reported 2164 cases of dengue fever, corresponding to a rate of 31.9 cases per 100,000 population. Among these cases, 2.4% (52 cases) were classified as severe, translating to a rate of 0.8 per 100,000 population. In the Urabá region, 956 cases of severe disease were documented, representing a rate of 179.1 cases per 100,000 population. Significant occurrences were observed in the municipalities of Apartadó (31.9%), Turbo (24.9%), and Chigorodó (21.3%) [15]. Within the department of Antioquia, dengue cases have shown a significant decrease among adults, but have increased among children and young people between 2017 and 2023, indicating a shift in the epidemiological profile [14]. These findings suggest a trend towards more severe clinical presentations, signaling an epidemiological shift in the disease pattern.

A multitude of studies have investigated risk factors that may signal the progression of dengue from dengue without warning signs (DWOS) or dengue with warning signs (DWS) to severe dengue (SD) [9,16,17,18,19]. Factors such as a history of cardiovascular diseases, diabetes, and specific gene polymorphisms associated with immune response have been identified as potentially increasing susceptibility to severe dengue [16,17]. Dehydration during the critical phase is also crucial: delayed diagnosis can exacerbate the severity, leading to complications like severe bleeding [16,17]. Other associated risk factors include age, dengue NS1 antigenemia, and the simultaneous presence of anti-dengue IgM and IgG antibodies; however, indicators of disease severity are not consistently uniform across various populations and regions [9,16]. These variations may be influenced by multiple factors, including geographical location, parameters of Aedes mosquitoes, population age, and the co-circulation of different virus genotypes [18,19,20].

While predictive models have been beneficial in public health, many rely on basic regression or cross-sectional data and often lack comprehensive validation [9,16,21]. The advent of contemporary machine learning techniques, such as neural networks and decision trees, has enabled the rapid development of predictive models in healthcare [21,22]. These models have shown promise in forecasting dengue outbreaks and identifying their locations, but their use in assessing disease severity is still limited [23,24]. Utilizing these advanced analytical tools could significantly enhance the prediction of severe dengue risk.

The timely identification of individuals at high risk for severe dengue is essential for decreasing case fatality rates, optimizing resource allocation, and improving clinical management protocols. Acknowledging this necessity, particularly in the context of significant dengue outbreaks, this study seeks to develop a predictive model for severe dengue, using data from Urabá, a hyperendemic region.

## 2. Materials and Methods

### 2.1. Participants

The participants were from Apartadó, Turbo, and Chigorodó, three of the eleven towns in the Urabá region. Situated within the department of Antioquia, Colombia, the Urabá region covers an area of 11,664 km^2^, representing 18.6% of Antioquia’s territory. These municipalities experience a tropical climate with temperatures ranging from 23 °C to 40 °C. According to 2022 predictions, the population of Antioquia is estimated to be 6,787,846, with 533,768 residing in the Urabá region. Among this population, 59.2% live in urban areas, while 217,836 people reside in rural regions [25]. This study was conducted in accordance with the principles outlined in the Declaration of Helsinki [26] and adhered to the guidelines of the Colombian Resolution 8430 of 1993 [27].

The Urabá region witnessed a marked increase in dengue incidence from 2015 to 2019, with a 67.2% rise in overall cases and an 88.9% increase in severe dengue cases [28]. In 2019, Apartadó recorded 555 dengue fever cases, Turbo 241 cases, and Chigorodó 137 cases, more than doubling the figures from 2015. This underscores the region’s significance for investigating predictors of severe dengue [15].

A descriptive cross-sectional observational study was conducted, adhering to the STROBE checklist guidelines detailed in Appendix A. Figure 1 depicts the flow of the sample used for predictive analysis. This study analyzed data from 2018 to 2022, including 233 patients. Data were sourced primarily from two cohorts. The first source was a prospective cohort of newly identified cases from November 2020 to September 2022, which included 192 patients, 6 of whom had severe cases. Owing to the small number of severe cases in this cohort, a second data source was added, featuring 41 retrospective cases from two distinct periods: 2018 to 2019 and October to December 2022. Among the 47 patients diagnosed with severe dengue, symptoms included severe hemorrhage (*n* = 6), significant organ damage (*n* = 17), and extensive fluid leakage (*n* = 24).

Patients were recruited through active searches in both hospital databases and community settings. Trained nurses carried out epidemiological surveillance during outbreaks, visiting homes to connect with patients who had sought medical help. The cohort included individuals of all ages, genders, and ethnicities presenting with fever lasting up to 7 days, without a clear infectious source, and showing at least two of the following symptoms: headache, pain behind the eyes, muscle pain, joint pain, or skin rash. Confirmation of dengue was determined through specific laboratory tests, ensuring that all participants met the critical diagnostic criteria. Exclusion criteria included individuals who had received blood product transfusions within three months before the study or those with a history of blood disorders. Patients were assessed during their febrile phase and followed through the recovery phase until day 14 or 21, marking the resolution of symptoms.

Retrospective cases were identified by reviewing clinical records. The distribution of these cases included 11 from 2022, 4 from 2020, 3 from 2019, 22 from 2018, and 1 from 2016. It is important to note that these additional cases lacked the detailed information gathered for the original cohort, as they were not subjected to all the initial analyses.

Eligibility criteria for study participants required laboratory-confirmed dengue virus (DENV) infection, with the satisfaction of criteria of one of the following: (1) positive DENV-specific real-time reverse transcription polymerase chain reaction (RT-PCR); this test was conducted on all serum samples collected during the acute phase, following RNA extraction, (2) confirmation of the *Dengue virus* NS1 antigen using capture ELISA on acute-phase serum samples, or (3) seroconversion of IgM or IgG antibodies in paired samples [5].

The identification of viral RNA through RT-PCR was conducted utilizing a Centers for Disease Control and Prevention (CDC) dengue virus serotype detection kit, in adherence to the manufacturer’s guidelines. IgM antibody detection was assessed using the Panbio capture ELISA method, following the manufacturer’s protocol, across both acute and convalescent phase serum samples. IgG antibody detection was carried out using the focus capture ELISA method, following the manufacturer’s instructions. This process was applied to all acute-phase serum samples, with repeat testing during the convalescent phase 14 days later in cases of initial negative outcomes.

### 2.2. Case Classification

Cases were categorized according to the WHO’s 2009 dengue guidelines [5] into two groups: non-severe dengue (non-SD) and severe dengue (SD). A case was classified as SD if it met any of the following criteria: (a) evidence of shock or accumulation of pleural/peritoneal fluid leading to respiratory failure, (b) severe bleeding requiring medical intervention, (c) significant impairment of vital organs, such as acute liver failure (defined as AST and ALT levels ≥ 1000 U/L), acute kidney injury (AKI) with a serum creatinine level increase of ≥0.3 mg/dL within 48 h or an elevation of ≥1.5 times from baseline within 7 days, encephalopathy (evidenced by seizures or disturbances in consciousness), or signs of myocarditis or heart failure.

### 2.3. Data Collection

For the cohort study, the primary source of information was the patients themselves, supplemented by data extracted from their medical records after receiving healthcare. A detailed, standardized clinical data collection form was designed to classify laboratory-confirmed cases of dengue. Another key data source was the hospital’s electronic medical records system, which offered a rich array of socio-demographic information, including age, gender, ethnicity, education, occupation, socioeconomic status, type of insurance, and place of origin. Clinical data included the type of disease diagnosed; for instance, the presence of both IgG and IgM antibodies indicates a secondary infection, comorbidities, healthcare history, presenting symptoms, and laboratory metrics such as blood counts and liver and renal function tests. The timeline of symptom onset, initial consultation, and the emergence of disease severity was meticulously documented.

To effectively prognosticate the disease trajectory, our focus was primarily on clinical and laboratory information obtained within the first five days following symptom onset, corresponding to the phase of the febrile disease. Indeed, some variables were not extracted on the same day; nevertheless, all of them pertain to the febrile phase of the disease. This approach highlighted the critical importance of initial clinical evaluations and primary laboratory findings following the patient’s first consultation. Data from this initial phase were utilized as potential predictors of severity in our predictive model. Furthermore, we also investigated potential co-infections with other pathogens, including Leptospira, Zika, and Chikungunya. The Panbio ELISA was employed for detecting Leptospirosis. For Chikungunya, IgM antibodies were identified using the Novatec ELISA, and viral RNA was detected via RT-PCR utilizing the CDC Trioplex diagnostic kit. The same RT-PCR method with the CDC Trioplex kit was also used for detecting Zika virus RNA. In alignment with the protocol for monitoring febrile diseases in Urabá Antioqueño, all patients underwent malaria testing.

### 2.4. Statistical Analysis

To identify early indicators of severe dengue (SD), we analyzed clinical data from initial hospital presentations prior to the development of SD. Comparisons between the SD and non-SD groups were made using Fisher’s tests for qualitative variables and Mann–Whitney U tests for quantitative variables. A *p*-value below 0.05 was considered statistically significant.

Multiple correspondence analysis (MCA) was used to identify patient profiles based on clinical manifestations and outcomes, focusing on distinguishing between SD and non-SD cohorts. MCA, which is effective in reducing the dimensionality of nominal categorical data, positions these data in a factorial space using the chi-squared distance to outline relationships between categories. A 2D scatter plot, illustrating the first two principal components, was used for visual interpretation, grouping similar profiles and distinguishing between SD and non-SD patient archetypes.

During the data analysis phase, we encountered inconsistencies in the form of missing data, especially in sections related to laboratory tests, hemograms, and liver function tests. To address these gaps, a rigorous data imputation technique was employed. Missing values were replaced with the average of the available data, calculated with precision. To enhance accuracy, this average was computed using data from individuals of the same gender and within a five-year age range of the patient with missing data. Notably, approximately 25% of the liver function test results required imputation, and less than 5% of hemoleucogram values had missing data that needed attention. This method ensured the accuracy and clinical relevance of the imputations, preserving the integrity and credibility of our dataset while minimizing potential biases.

Three machine learning models—logistic regression, decision tree classifier, and random forest classifier—were assessed through 10-fold cross-validation of the training data. The logistic regression model demonstrated a mean accuracy of 0.89 with a standard deviation of 0.08, outperforming the other models tested. The support vector machine reported a mean accuracy of 0.87 and a standard deviation of 0.06. The decision tree model had a mean accuracy of 0.82 with a standard deviation of 0.07, and the random forest model showed a mean accuracy of 0.89, but with a lower standard deviation of 0.04 compared to logistic regression. Despite the similar mean accuracy values between logistic regression and random forest, logistic regression was chosen for further optimization due to its higher interpretability, which is crucial for clinical applications. Hyperparameter tuning was performed via grid search, examining various parameters including regularization strengths (C), penalty types (l1, l2, and ElasticNet), the mix of l1 and l2 in ElasticNet, and the optimization algorithm (saga). The optimal parameters were identified after conducting another round of 10-fold cross-validation.

Logistic Regression is advantageous as it allows the use of regularization techniques to prevent overfitting, a common challenge in predictive modeling. In this study, the Grid Search method was employed to systematically explore a range of hyperparameters to find the optimal combination, thereby enhancing the model’s accuracy and generalizability. Specifically, the term ‘ElasticNet’ refers to a type of regularization used during this optimization process. ElasticNet is a hybrid approach that combines the strengths of L1 and L2 regularization methods. This blend is particularly beneficial for dealing with correlated predictors as it helps in balancing variable selection and model complexity, effectively preventing overfitting. Together, these techniques ensure the development of a robust model that not only reliably predicts outcomes, but also maintains clarity and simplicity, preventing the model from becoming overly complex.

Prior to training, Z-score normalization was applied to ensure uniform variable contribution. The dataset was split into an 80% training subset and a 20% testing subset. Post-hyperparameter tuning, the model’s performance was assessed on both subsets using metrics like accuracy, precision, recall, F1 score, ROC curve, and AUC with its 95% confidence interval (IC). AUC benchmarks were classified as follows: 0.90–1 (excellent), 0.80–0.90 (good), 0.70–0.80 (fair), 0.60–0.70 (below average), and 0.50–0.60 (inadequate) [29].

Before finalizing the logistic regression model’s conclusions, foundational assumptions were validated. This included verifying the linear relationship between independent variables and log-odds, assessing multicollinearity using the variance inflation factor (VIF), identifying significant outliers and influential points using residual plots and Cook’s distance, and evaluating model fit using the Hosmer–Lemeshow test. A non-significant result (*p* > 0.05) indicated an appropriate model fit. The model was built using variables that demonstrated statistical significance in the bivariate analysis. The exclusion process considered the restrictions of the predictive model based on the number of events per predictor and clinical relevance. A key criterion for excluding variables from the final model was high collinearity between predictors (e.g., only one liver function test was retained from those available).

To assess the impact of individual predictors, odds ratios (ORs) were calculated using a generalized linear model (GLM) with a logit link function. ORs were derived directly from the coefficients of the predictors. To ensure the robustness of these estimates, a bootstrapping methodology with 1000 repetitions was employed to generate 95% confidence intervals (CIs). All statistical analyses were conducted using Python 3.11.5 and R 4.3.1, which provide comprehensive computational toolkits for the statistical and machine learning evaluations integral to this study. This dual-software approach leverages the strengths of both environments, enhancing the analytical capabilities necessary for rigorous data analysis.

## 3. Results

In the study, data from 233 patients, all confirmed with DENV infections, were analyzed. The median age was 10 years, with an interquartile range of 11 years. Of these, 47 progressed to severe dengue (SD) cases (28 males and 19 females), and 186 to non-severe dengue (non-SD) cases (105 males and 81 females), according to the WHO 2009 criteria. Individuals residing in rural areas were 77% less likely to belong to the SD group than the no SD group. The odds of SD in individuals with at least one comorbidity were almost three times the odds of no SD in individuals with at least one comorbidity, with an OR of 4.22 (*p*-value = 0.001). Although secondary infections were more frequent in no SD (89.2%) than in SD (72.1%), they were significantly associated with SD (OR: 3.21; *p* = 0.012). Detailed demographic and clinical characteristics of these patients are outlined in Table 1.

Various symptoms were significantly associated with disease severity, as indicated in Table 2, with a *p*-value of less than 0.05. The results showed an increased likelihood of severe dengue (SD) for symptoms such as jaundice, fluid accumulation, irritability, convulsions, hepatomegaly, splenomegaly, hematemesis, abdominal pain, nausea, edema, and ascites, each showing an odds ratio (OR) greater than 1.0. Conversely, symptoms such as retro-orbital pain, fatigue, dizziness, myalgia, arthralgia, exanthem, and pruritus were observed less frequently in patients with severe dengue, each with an OR less than 1.0. Appendix A clarifies that there are no significant variables among the additional studied factors.

Adjusted multiple correspondence analysis (MCA) highlighted that symptoms like irritability and organ enlargement (e.g., spleen, liver) were indicative of SD. Conversely, the absence of these symptoms, combined with the lack of dizziness and the presence of rashes, were characteristic of the non-SD group (see Figure 2).

Table 3 illustrates the differences in various hemogram parameters and other quantitative variables between patients with severe dengue (SD) and those with non-severe dengue (non-SD). Significant variations in median values based on disease severity were noted. In the non-SD group, higher median values were observed for respiratory rate, lymphocytes, and eosinophils. Conversely, the SD group showed higher median values for hemoglobin, hematocrit, white blood cells (WBC), neutrophils, aspartate aminotransferase (AST), alanine aminotransferase (ALT), C-reactive protein, creatinine, blood urea nitrogen, prothrombin time, and activated partial thromboplastin time; in contrast, the median platelet count was lower in the SD group (*p*-value < 0.05).

The refined predictive model, as detailed in Table 4, identified several key indicators associated with an increased likelihood of severe dengue (SD). Elevated leukocyte counts were linked to a nearly twofold increased risk of severe dengue (adjusted odds ratio, aOR = 1.87; 95% confidence interval, CI, 1.10–3.16). Similarly, higher neutrophil counts were associated with more than a threefold increased risk (aOR = 3.26; 95% CI, 1.71–6.23), and a higher liver function score was associated with a more than fourfold increase in the likelihood of severe dengue (aOR = 4.60; 95% CI, 1.68–12.63). Conversely, higher platelet counts were linked to an 82% reduced likelihood of developing severe dengue (aOR = 0.18; 95% CI, 0.08–0.43). Additionally, primary infections were found to decrease the risk by 55% compared to secondary infections (aOR = 0.45; 95% CI, 0.28–0.72).

The predictive model, incorporating six pivotal predictors, proficiently differentiated between SD and non-SD cases, registering an AUC of 0.91 (95% CI, 0.85–0.96), as seen in Figure 3.

The model exhibited a training accuracy of 0.89, which slightly increased to 0.91 in the testing phase, indicating robust calibration. In the training phase, performance metrics for non-severe cases—accuracy, recall, and F1 score—consistently exceeded 90%. For SD cases, while recall was 57%, accuracy reached 84%. During the testing phase, performance metrics for non-severe cases remained strong, exceeding 0.92, demonstrating consistent case categorization. Specifically, for SD cases in the test sample, precision, sensitivity, and F1 score were 88.0%, 70.0%, and 78.0%, respectively (see Table 5).

## 4. Discussion

This study revealed that patients with severe dengue (SD) frequently exhibited symptoms like irritability, fluid accumulation, abdominal pain, hepatomegaly, hematemesis, and enlargement of organs such as the spleen and liver. In contrast, patients with non-severe dengue often lacked these specific symptoms, but were more likely to display a rash and other milder manifestations. The multivariable predictive model identified secondary infections, along with platelet, leukocyte, neutrophil, and AST levels, as significant predictors of SD.

The increasing severity of dengue, particularly in Latin America, has been a major concern, with countries like Colombia, Peru, and Brazil witnessing significant surges [12,13]. The year 2019 alone saw over three million dengue fever cases in Latin American nations [30]. This escalation in incidence, coupled with a changing epidemiological landscape [31,32,33] and a rise in severe manifestations, underscores the need for improved hospital infrastructure [34]. This study presents a predictive model that aims to address the critical needs in patient triage and resource allocation, especially pertinent in Latin America where similar research is limited. This gap in research is highlighted by a 2023 review that identified one relevant study from Venezuela in 2012 [9], and a 2021 review citing just a Nicaraguan study [35].

In Colombia, machine learning techniques have been utilized to identify genes that improve the clinical prediction of dengue [36]. Additionally, the country has introduced web-based tools designed to assist in vector surveillance and entomological reporting in specific regions [37]. In Cali, a clinical implementation took place in 2019, resulting in the development of four algorithms: two based on signs and symptoms and two incorporating leukocyte count (≤4500/mm^3^) or platelet count (≤60,000/mm^3^). The algorithm that included hemogram parameters was the most accurate, showing a sensitivity of 76.5% (95% CI 71.9–80.5) and a specificity of 46.0% (95% CI 37.6–54.7). However, in external validation—that is, in the application to different populations that did not participate in the initial development of the model—its sensitivity dropped to 11.1% (95% CI 4.9–20.7), while its specificity rose to 91.9% (95% CI 87.5–93.9) [38]. Despite these advancements, the effective deployment of clinical support tools for dengue remains challenging.

These findings underscore the significance of neutrophils, leukocytes, platelets, and aspartate aminotransferase (AST) as dependable markers for identifying severe dengue. The model, boasting an area under the curve (AUC) of 0.91, demonstrates a high level of predictive accuracy. These markers, along with hematocrit, white cell count, and age, are important for early detection of the disease. The timely identification of alarming cases is pivotal in mitigating disease severity and mortality [9]. In this study, not all predictors were evident, but we did find that patients with severe dengue (SD) exhibited higher median levels of hematocrit and white blood cells (WBC). Laboratory indicators such as hypoalbuminemia, hypoproteinemia, and leukocytosis are particularly significant in predicting SD, although they apparently differ according to age [9,39,40]. Although leukocyte counts in this study remained within the normal range, they were statistically higher in SD cases compared to the non-SD group.

Interestingly, our study did not find age to be a significant factor in the severity of dengue, which may be due to the specific composition of our sample, encompassing individuals aged between 0 and 84 years. However, the median age was similar for both severe dengue (SD) and non-severe dengue (non-SD) groups (median age = 10 years). The presence of cases in younger groups could be related to the availability of a susceptible population. Among adults, there may be a higher prevalence of secondary infection, in addition to the presence of all four serotypes [41]. Furthermore, this phenomenon has tended to follow a pattern in recent years in the Americas, where a higher number of cases have been reported in individuals under 15 years of age [32,42]. Finally, the study period coincides with the COVID-19 pandemic, during which cases in adults may have been underreported, or the population may have avoided medical services due to fear of infection [42].

Given the role of hemodynamic predictors in our model, it is crucial to acknowledge factors that may influence the hematological and cardiovascular stability in dengue patients. Maintaining cardiac function is essential for hemodynamic equilibrium, and previous studies have highlighted dengue-related cardiac complications [16,17,43]. Recent research suggests a potential role of neutrophils, particularly myeloperoxidase (MPO), in these cardiac events [43]. Our results align with this theory, indicating a 2.3-fold increase in neutrophil counts in cases of severe dengue.

A 2021 meta-analysis of 122 studies identified a high platelet count as a protective factor against dengue severity (DS) [16]. This aligns with our observations and the WHO’s warning signs [5]. Likewise, leukocyte count has emerged as a critical determinant in our study; an elevated leukocyte count was associated with increased disease severity (aOR: 1.87). This marker is crucial for distinguishing between bacterial and viral infections, as well as differentiating DS from non-SD. Other studies have also noted the presence of atypical lymphocytes and immature platelets as indicators of severe dengue [44]. PAHO guidelines recognize leukocytosis as a predictor of dengue severity [7]. Other prognostic studies also converge on this finding [9,16].

Our findings emphasize serum biomarkers, especially aspartate aminotransferase (AST) and alanine aminotransferase (ALT), as highly predictive for severe dengue (SD). This is coherent with the literature, although some studies have shown that their predictive power diminishes in adults. [16,45]. It is noteworthy that C-reactive protein (CRP) levels were significantly elevated in cases of SD, corroborating observations made by other researchers in patients in intensive care units (ICU). Previous studies have shown that rapid changes in platelet count and AST levels within the first 72 h of fever onset can predict severe dengue [35]. In this study, AST emerged as a significant risk factor (aOR: 4.60), underscoring the importance of monitoring these markers during the febrile phase [16]. The AST/ALT ratio, serum albumin, and bilirubin levels also have prognostic value for severe dengue, with elevated AST/ALT ratios linked to early mortality risks [46], highlighting liver dysfunction as a key feature in severe dengue episodes [44,47]. In fact, some new biomarkers of liver function in dengue infection have reported that dengue-induced liver damage predominantly initiates in the centrilobular area and progresses toward the portal area during dengue infection [47].

The analysis revealed that primary infection (aOR = 0.45) serves as a protective factor compared to secondary infection. Serotype-specific immune responses are typically strong, but subsequent exposure to a different serotype can trigger antibody-dependent enhancement (ADE), where non-neutralizing antibodies facilitate viral entry into host cells, potentially increasing disease severity [48]. This underscores the complex role of antibody responses in dengue pathology. Indeed, the literature frequently associates serotype and infection type (primary vs. secondary) with severe dengue cases [10,49].

The serotype of the initial infection (*DENV-1*, *DENV-2*, *DENV-3*, *DENV-4*), the sequence of subsequent infections, and the interval between them may significantly impact disease severity [50,51]. Notably, studies in Nicaragua, India, and Mexico [52] report an association between *DENV-3* and severe primary infections, highlighting the need for serotype-specific analyses. However, due to limitations in serotype data, further exploration of these interactions was not possible in this study. Future research integrating viral and immunological profiling could provide deeper insights into ADE’s role in dengue severity.

Other studies have focused on NS1 antigenemia, gender, and age as key predictors, along with mild increases in IgG and IgM concentrations related to disease severity [18]; the concept of triple positivity (NS1, IgM, IgG) as an indicator of severe dengue has been suggested [44]. Moreover, secondary infection has been consistently associated with increased severity [16,44]. While these predictors are valuable, significant advancements in early-stage diagnostics (1 to 4 days post-infection) are needed for truly transformative impact in dengue management [53].

Prognostic studies in dengue are still at an early stage, with many lacking the rigorous statistical methods required for accurately assessing the sensitivity and specificity of proposed indicators [9,35,44]. Despite its limitations, this study marks a significant advancement in the field. The study presents a model with good accuracy and a well-balanced sensitivity and specificity across severity groups (SD vs. non-SD) in both training and validation phases. The analytical approach and precise calibration underpin the strength of the research. Moreover, given the scarcity of such detailed research in Latin America, and specifically in Colombia [9], our study is important for its contemporary relevance and methodological rigor in developing predictors for dengue severity.

Furthermore, with the coordination and support of the country’s health system, along with the provision of resources for liver testing, the model identified in this study could be uniformly applied in primary care settings. Therefore, its ultimate impact could be crucial in controlling severe disease. The utilization of predictive models in primary clinical settings has shown significant contributions, as evidenced by previous implementation studies in other countries. The management and identification of cases, as well as their differential classification in primary and secondary health services, are crucial determinants in the clinical outcome of dengue [54,55]. In this context, Colombia’s Early Warning and Response System (EWARS) plays a key role by integrating climatic and epidemiological data to anticipate dengue outbreaks and guide timely interventions. Despite its effectiveness, challenges remain in enhancing real-time data quality and incorporating socio-economic factors to improve prediction accuracy [56]. In response to this need, predictive models have emerged, and the present study is linked to them. Research contributes to resolving the uncertainties surrounding the progression of dengue disease and enhances the health system’s ability to anticipate whether a patient will experience a mild manifestation or progress to a more severe, potentially fatal one [57].

The predictive model developed in this study has significant clinical applications, particularly in low-complexity and rural settings where access to specialized care is limited [41,58]. By relying on routine laboratory markers, it enables early identification of severe dengue (SD) risk, supporting timely triage, optimized resource allocation, and improved patient referral. This is especially crucial during outbreaks, helping frontline healthcare providers prioritize high-risk cases and reduce the burden on tertiary hospitals. Future validation in real-world settings will further assess its feasibility and impact on patient outcomes.

Despite the existence of significant studies on the prediction of dengue severity [16,35], their implementation in real clinical settings is limited, especially in Colombia, where the use of predictive models has only recently begun [37,38]. Global initiatives in India and Asia have demonstrated the contributions of digital tools, software, and applications in this field [37,59,60,61,62]. Although this study does not develop such a tool, it establishes the foundations for the predictors that should be considered and serves as a starting point for predictive models and systems applied in the clinical context.

This study is not without its limitations. The challenges we faced were closely tied to the broader healthcare context of the period studied. While commitment to using complete data records strengthened the authenticity and reliability of the findings, it also limited the sample size, potentially affecting the generalizability of our conclusions. A higher number of cases was expected given that this is a hyperendemic area. However, the dynamics of other diseases, such as COVID-19, malaria, may have led to an underestimation of cases or even their presumptive treatment as other febrile illnesses. Nevertheless, the researchers conducted active hospital and community-based surveillance, along with a thorough review of medical records, to ensure the inclusion of all suspected dengue cases. Furthermore, the complex clinical manifestations of SD, particularly hemodynamic shifts, may interact in intricate ways with various prognostic factors, adding complexity to disease progression assessments.

In our study, we focused on routinely available clinical and laboratory parameters to enhance the practical applicability of dengue severity prediction in real-world healthcare settings. While factors such as viral serotype, viral load, and host immune response are known to influence disease severity, these data were not consistently available, as they are not routinely recorded in medical records or standard clinical practice in many healthcare facilities. Additionally, the retrospective nature of a portion of study and the reliance on secondary sources limited access to virological and immunological data. Future prospective studies incorporating these variables could provide further insights into their role in improving dengue severity prediction models.

Another aspect that significantly impacted the study is the reliance on secondary data (41 cases of severe dengue), which has implications such as the lack of data on key variables like serotypes, types of infection, and liver function. In some instances, the original variables were retained, while in others, where significant data loss occurred, imputation was employed. Although this was performed in a controlled manner, the absence of data may affect the findings. Future research could explore hybrid models that integrate traditional epidemiological methods with machine learning to improve dengue severity prediction. This approach may enhance accuracy while maintaining interpretability, especially with high-quality prospective data. Some applications of this type have demonstrated important results [63,64].

The temporal scope of study, especially from 2020 onwards, intersected with the COVID-19 pandemic, presenting unique challenges. For instance, the data from 2020 to 2022 recorded 213 dengue cases, but this number may not accurately reflect the true prevalence due to pandemic-related disruptions. Symptomatic similarities between dengue and COVID-19, along with the heightened focus on the latter, could have led to underreporting of dengue cases. Misdiagnoses due to COVID-19 concerns and patients’ reluctance to seek medical care due to fear of COVID-19 exposure might have resulted in underreported dengue cases. The inclusion of data from 2018 and 2019 helped mitigate this issue to some extent, but the impact of the pandemic on dataset cannot be ignored.

## 5. Conclusions

In conclusion, the present study advocates for integrating AST monitoring with the established 2009 WHO guideline warning signs—neutrophils, platelets, and leukocytes—to enhance predictions of SD risk. Incorporating insights into viral serotype and immunological history (distinguishing between primary and secondary infections) could further refine these risk assessments. The model, a good contribution in Latin America and particularly in Colombia, equips healthcare professionals with a valuable tool for rapidly assessing and prioritizing severe dengue risks, especially during outbreaks. Its reliance on initial diagnostic data enhances its applicability across various healthcare settings, from tertiary hospitals to community clinics.

## Figures and Tables

**Figure 1 viruses-17-00740-f001:**
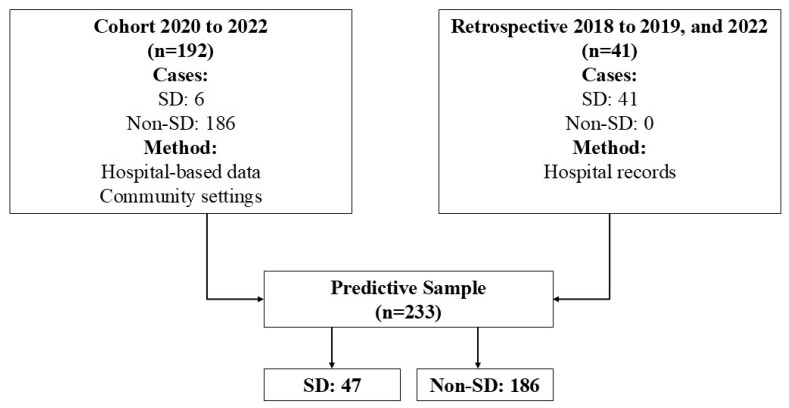
Flowchart patients in analysis predictive. Note: SD: severe dengue.

**Figure 2 viruses-17-00740-f002:**
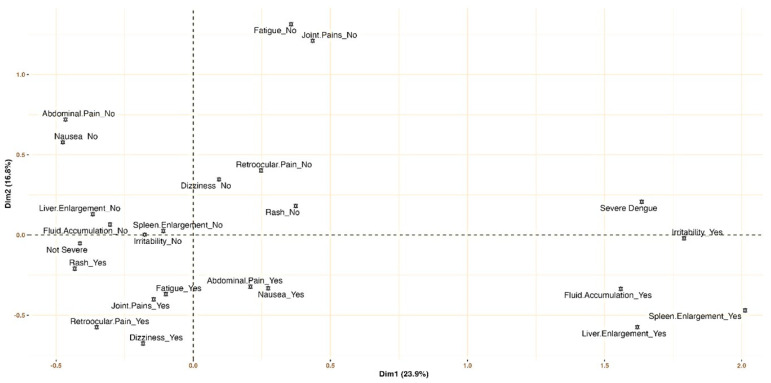
Signs and symptoms profiling for severe dengue and non-severe dengue: multiple correspondence analysis.

**Figure 3 viruses-17-00740-f003:**
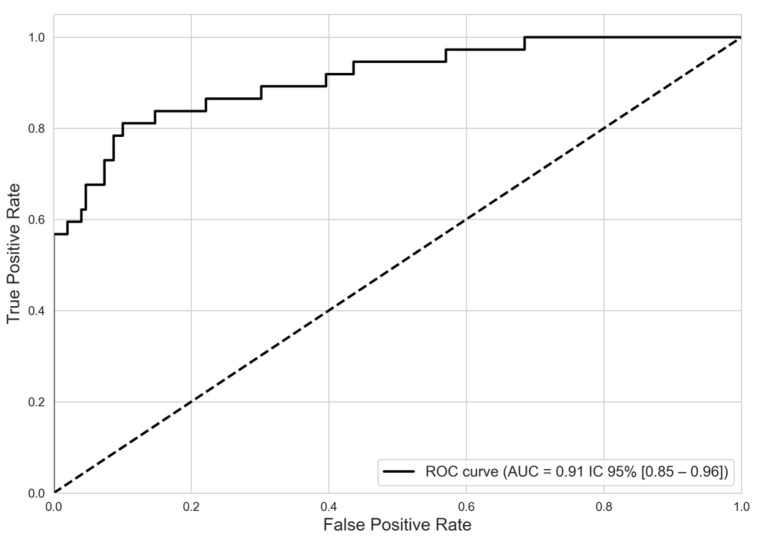
Discriminatory performance of model predictive severe dengue from non-severe dengue. ROC curve. NOTE. ROC: receiver operating characteristic. AUC: area under the curve.

**Table 1 viruses-17-00740-t001:** Characteristics and outcomes for dengue patients.

Variable	SD		No SD		OR	*p*-Value ^a^
	*n* = 47	%	*n* = 186	%		
Age						
Mean/std	18.3/18.7		13.1/11.8		Na	0.293 ^b^
Median/IR	10.0/16.0		10.9/9.0			
Sex						
Male	28	59.6	105	56.5	1.14	0.829
Female	19	40.4	81	43.5		
Place of residence						
Rural	7	14.9	81	43.5	0.23	<0.001
Urban	40	85.1	105	56.5		
Health Insurance type						
Contributory	33	70.2	104	55.9	1.86	0.104
Subsidized	14	29.8	82	44.1		
Comorbidity						
Yes	14	29.8	17	9.1	4.22	0.001
No	33	70.2	169	90.9		
NS1 (*n* = 203) *						
Negative	3	17.6	77	41.4	0.30	0.097
Positive	14	82.4	109	58.6		
IgM (*n* = 229) *						
Negative	5	11.6	46	24.7	0.40	0.097
Positive	38	88.4	140	75.3		
IgG (*n* = 227) *						
Negative	8	19.5	20	10.8	2.01	0.200
Positive	33	80.5	166	89.2		
RT-PCR (*n* = 194) *						
Negative	3	37.5	93	50.0	0.60	0.740
Positive	5	62.5	93	50.0		
Infection *						
Secondary	31	72.1	166	89.2	3.21	0.012
Primary	12	27.9	20	10.8		
Serotype *						
*DENV-1*	2	50.0	64	68.8	0.45	0.765
*DENV-2*	2	50.0	29	31.2		
Co-infection						
Yes	7	14.9	37	19.9	1.42	0.579
No	40	85.1	149	80.1		
Malaria						
Yes	5	10.6	2	1.1	10.95	0.003
No	42	89.4	184	98.9		
Leptospirosis						
Yes	2	4.3	35	18.8	0.19	0.027
No	45	95.7	151	81.2		

NOTE. SD: severe dengue. ^a^: Fisher test. ^b^: Mann–Whitney U test. IR: interquartile range. Std: standard deviation. * missing NS1: 30 data. IgM. IgG missing: 6. RT-PCR missing: 39. Infection missing: 4. Serotype missing: 136.

**Table 2 viruses-17-00740-t002:** Differences in signs and symptoms by SD versus non-DS.

Variable	SD		Non-SD		OR	*p*-Value ^a^
	*n* = 47	%	*n* = 186	%		
Retro-orbital pain	8	17.0	88	45.4	0.23	<0.001
Abdominal pain	40	85.1	121	53.5	3.07	0.013
Nausea	39	83.0	109	48.4	3.44	0.003
Fatigue	31	66.0	151	69.6	0.45	0.040
Dizziness	8	17.0	71	36.6	0.33	0.010
Myalgia	25	53.2	154	73.0	0.24	<0.001
Arthralgia	26	55.3	149	70.3	0.31	0.001
Exanthem	8	17.0	100	51.5	0.18	<0.001
Pruritus	5	10.6	92	48.2	0.12	<0.001
Jaundice	5	10.6	3	1.6	7.26	0.010
Edema of extremities	11	23.4	16	8.1	3.25	0.010
Fluid accumulation	24	51.1	14	6.7	12.82	<0.001
Ascites	11	23.4	10	5.1	5.38	<0.001
Respiratory distress	5	10.6	0	0.0	-	<0.001
Irritability	16	34.0	5	2.5	18.68	<0.001
Convulsion	5	10.6	1	0.5	22.02	0.001
Hepatomegaly	27	57.4	15	7.0	16.20	<0.001
Splenomegaly	8	17.0	4	2.1	9.33	<0.001
Hematemesis	16	34.0	4	2.0	23.48	<0.001

NOTE. ^a^: Fisher test. SD: severe dengue. Not data mining for variables.

**Table 3 viruses-17-00740-t003:** Quantitative variables by SD versus non-DS.

Variable	SD		Non-SD		*p*-Value ^a^
	Min–Max	Median [Q1–Q3]	Min–Max	Median [Q1–Q3]	
Age (years completed)	0.4–84.0	10.0 [6.0–22.0]	0.6–70.0	10.0 [6.0–15.0]	0.293
Systolic blood pressure (mm/Hg)	70.0–172.0	101.0 [97.3–111.8]	85.0–143.0	100.0 [95.0–110.0]	0.283
Diastolic blood pressure (mmHg)	40.0–91.0	63.0 [60.0–71.0]	40.0–94.0	61.0 [60.0–70.0]	0.856
Mean arterial pressure (mmHg)	50.0–140.0	80.0 [73.7–87.7]	56.3–109.0	74.7 [71.4–83.3]	0.050
Pulse pressure (mmHg)	20.0–81.0	40.0 [36.0–46.0]	10.0–68.0	40.0 [35.0–40.0]	0.250
Heart rate (pulse for one minute)	68.0–140.0	105.0 [89.0–116.4]	50.0–151.0	98.0 [88.0–111.8]	0.160
Respiratory rate (number of breaths for one minute)	14.0–40.0	20.0 [18.3–23.0]	16.0–45.0	21.5 [20.0–24.0]	0.041
Oxygen saturation (%)	56.0–99.0	98.0 [97.0–99.0]	90.0–99.0	98.0 [98.0–99.0]	0.113
Hemoglobin (gr/dL)	7.9–18.6	13.1 [11.9–14.6]	5.1–19.4	12.4 [11.5–13.5]	0.032
Hematocrit (%)	23.9–63.1	40.5 [34.7–44.7]	15.3–57.6	37.3 [34.6–40.6]	0.025
Platelets (mm^3^)	7000.0–422,000.0	60,000.0 [31,000.0–109,500.0]	5200.0–534,000.0	147,381.6 [93,500.0–215,500.0]	0.000
White blood cells (mm^3^)	1600.0–23,000.0	6414.0 [4005.0–9650.0]	1437.0–45,000.0	4425.0 [3302.5–6165.0]	0.000
Neutrophils (%)	13.0–95.2	56.0 [47.9–73.5]	8.6–92.0	50.0 [36.0–65.4]	0.008
Lymphocytes (%) ^b^	0.3–77.0	31.2 [17.5–38.9]	3.9–81.2	36.1 [24.0–46.6]	0.009
Eosinophils (%) ^c^	0.0–9.0	0.7 [0.14–1.0]	0.0–19.7	2.4 [0.61–4.71]	0.000
Basophils (%) ^d^	0.0–4.0	0.9 [0.7–1.7]	0.0–5.0	1.0 [0.61–1.21]	0.787
Aspartate Aminotransferase (UI/L) ^e^	13.0–3000.0	133.1 [71.6–620.5]	15.3–469.6	94.6 [61.0–114.0]	0.000
Alanine Aminotransferase (UI/L) ^e^	11.0–3000.0	63.5 [35.0–229.0]	9.5–481.9	48.3 [28.8–65.2]	0.001
C-reactive protein (mg/dL) ^f^	0.2–276.0	4.5 [1.31–12.4]	0.0–96.9	1.0 [0.3–3.6]	0.008
Creatinine (UI/L) ^g^	0.3–9.6	0.7 [0.6–1.0]	0.2–2.6	0.5 [0.4–0.7]	0.000
Blood urea nitrogen (mg/dL) ^h^	4.1–94.2	14.5 [10.0–28.0]	2.0–81.4	9.2 [8.0–11.3]	0.000
Prothrombin time (seconds) ^i^	10.0–20.0	13.7 [11.5–16.7]	10.0–20.0	12.8 [11.5–14.8]	0.030
Activated partial thromboplastin time (seconds) ^i^	27.6–70.0	41.8 [38.1–54.3]	24.9–68.0	36.7 [34.3–38.8]	0.000

NOTE. ^a^: Mann–Whitney U test. SD: severe dengue. Min: minimum. Max: maximum. ^b^: missing SD 1. ^c^: missing SD 2. ^d^: missing SD 5. ^e^: missing SD 2, non-SD 4. ^f^: missing SD 21, non-SD 103. ^g^: missing non-SD 8. ^h^: missing SD 3, non-SD 7. ^i^: missing SD 6, non-SD 5.

**Table 4 viruses-17-00740-t004:** Multivariable prediction that predicts severe dengue.

Clinical Parameter	Coefficient	SE	*p*-Value	aOR	95% CI	
(Intercept)	−2.782	0.526	<0.001			
Primary infection *	−0.794	0.239	0.001	0.452	0.283	0.723
Platelets	−1.706	0.434	<0.001	0.182	0.078	0.425
Leukocytes	0.623	0.269	0.021	1.865	1.101	3.160
Eosinophils	−1.305	0.657	0.047	0.271	0.075	0.982
Neutrophils	1.181	0.330	0.000	3.258	1.705	6.227
AST (liver function test)	1.526	0.515	0.003	4.600	1.675	12.632

NOTE. SE: standard error. aOR: adjusted odds ratio. CI: confidence interval. * Reference category: secondary infection. AST: aspartate aminotransferase.

**Table 5 viruses-17-00740-t005:** Indicators of fit of the predictive model for severe dengue fever.

	Training ^a^		Test ^b^	
Metrics/Dataset	Non-SD	SD	Non-SD	SD
Precision	0.90	0.84	0.92	0.88
Recall (sensitivity)	0.97	0.57	0.97	0.70
F1 score	0.94	0.68	0.95	0.78
Support	149	37	37	10

NOTE. SD: severe dengue. ^a^: metrics of dataset training. Accuracy: 0.89; macro avg F1 score: 0.81; weighted avg F1 score: 0.88. ^b^: metrics of dataset test. Accuracy: 0.91; macro avg F1 score: 0.86; weighted avg F1 score: 0.91.

## Data Availability

The data used in this study are publicly available. The raw and processed data used for the analyses are available at https://github.com/jemilios/Dengue-Severity-Prediction-in-Colombia (accessed on 1 June 2024). The data on dengue severity prediction can be found in the respective files within the repository.

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
