# Peer review of "Dengue Severity Prediction in a Hyperendemic Region in Colombia"

_viruses, 2025, doi:10.3390/v17060740_

Round 1
Reviewer 1 Report
Comments and Suggestions for Authors
We consider the study relevant and innovative in terms of public health. However, I would like the authors to correct some points.
Figure 1 could be improved. The lines do not mach in soome parts. You can use a smaller font. Remove the (-) in the figure.
In methods, how much criterias were used to consider the samples positive? Criterion 3, IgM or IgG seroconversion can not be used isolate. A patient who is only IgG positive is not necessarily is positive for dengue at the moment.
In line 290, wouldn't it be the opposite? No-SD presentes 89.2% in relation SD.
In discussion, line 397, the authors explaind teh median age, around10 years, but they do not discuss the reason for this fact. It is interesting to note that this probably occurred due to the pandemic period where people did not seek hospital medical assistance for fear of COVID infection and this may impact your data of age.
Author Response
Figure 1 could be improved. The lines do not mach in soome parts. You can use a smaller font. Remove the (-) in the figure. The improved Figure 1 incorporates the suggested points. In methods, how much criterias were used to consider the samples positive? Criterion 3, IgM or IgG seroconversion can not be used isolate. A patient who is only IgG positive is not necessarily is positive for dengue at the moment. A sample was considered positive if it met at least one of the following: NS1 antigen detection, RT-PCR positivity, or IgM/IgG seroconversion in paired samples. We acknowledge that isolated IgG positivity does not confirm active infection, so we emphasize seroconversion for recent cases. This aligns with Colombian Ministry of Health and PAHO guidelines. In line 290, wouldn't it be the opposite? No-SD presentes 89.2% in relation SD. The writing was organized to facilitate comprehension. The results indicate an 89.2% secondary infection rate in patients without severe dengue. In discussion, line 397, the authors explaind teh median age, around 10 years, but they do not discuss the reason for this fact. It is interesting to note that this probably occurred due to the pandemic period where people did not seek hospital medical assistance for fear of COVID infection and this may impact your data of age. We agree that the COVID-19 pandemic may have influenced the median age observed in our study. We have now incorporated this point into the discussion and added additional context to acknowledge its potential impact on our findings.Reviewer 2 Report
Comments and Suggestions for Authors
This is an interesting manuscript that aims to identify predictors of severe dengue in a hyperendemicregion of Colombia.
I have some few observation regarding it:
As it is performed in a hyperendemic region, I would imagine a higher number of patients. Do the authors believe that a minor numerical difference between severe and non-severe dengue patients could provide evidence of other prevailing factors?
When describing participants (lines 117-128), they should fit better in introduction.
Do the authors know what serotype of dengue virus was positive in the patients?
Author Response
As it is performed in a hyperendemic region, I would imagine a higher number of patients. Do the authors believe that a minor numerical difference between severe and non-severe dengue patients could provide evidence of other prevailing factors?While a larger sample might be expected in a hyperendemic region, our study was limited to confirmed cases meeting strict inclusion criteria to ensure data quality. We actively search for institutional and community cases to ensure that all suspected cases are addressed. However, some aspects such as the covid-19 pandemic may overlap the existence of cases.
Despite the minor numerical difference between severe and non-severe cases, our analysis focused on identifying key predictive factors beyond case proportions. Statistical methods ensured that even small differences were assessed for clinical and epidemiological relevance.
These aspects were clarified in the discussion section.
When describing participants (lines 117-128), they should fit better in introduction.
Lines 80-89 provided context on the event in the region. However, the introduction emphasizes that these are the municipalities included in the study.
We decided to retain this information in the methodology while also specifying this aspect of the population at the end of the introduction. Do the authors know what serotype of dengue virus was positive in the patients? Yes, the dengue virus serotype was identified in the study cohorts. However, this information was not available for all patients from secondary sources. The detected serotypes were DENV-1 and DENV-2. Detailed information is provided in Table 1.
Reviewer 3 Report
Comments and Suggestions for Authors
The paper titled "Dengue Severity Prediction in a Hyperendemic Region in Colombia" by Jorge Emilio Salazar Flórez et al. provides a valuable contribution to predicting severe dengue in endemic regions. Its findings are particularly relevant during outbreaks.
I have the following comments and suggestions:
General Comments
The study concludes that monitoring liver enzymes, neutrophils, platelets, leukocytes, etc. improves the prediction of severe dengue risk. However, it does not address other critical risk factors, such as viral serotypes, viral load, and host immune responses, which could enhance prediction accuracy.
Specific Comments
- Study Sites and Sampling: The methods section needs further clarification on the rationale for site selection. It would be beneficial if the chosen sites were representative of Colombia as a whole. Additionally, the selection criteria for prospective and retrospective samples should be explicitly stated.
- Background: Briefly mentioning the clinical burden of severe dengue in Colombia would strengthen the study’s justification.
- Machine Learning Approach: Given that the authors acknowledge issues with retrospective data quality, is a machine learning approach appropriate for this setting? Would a hybrid model integrating traditional epidemiological methods with ML offer a better balance between accuracy and interpretability?
- Geographic Representation: Including maps of Colombia showing study sites would improve clarity and contextual understanding.
- Clinical Implications: The clinical significance of the findings should be emphasized. How do they enhance current triage protocols?
- Model Application in Clinical Settings: Instead of broadly stating that the model is useful for triage, specify how it could be integrated into clinical workflows.
- Secondary Infection and Antibody-Dependent Enhancement (ADE): The study notes that secondary infections are associated with severe dengue. Could the authors elaborate on whether this is linked to ADE? For reference, DENV-3 in Nicaragua has been associated with more severe primary infections, as well as in India and Mexico (Lancet, October 25, 2024).
Minor Suggestions
- Figure 2: The figure is difficult to interpret. Consider providing a clearer explanation in the legend or removing it if it does not add substantial value.
- The discussion section could be more concise. Some parts are overly detailed and could be streamlined for better readability.
- The current early warning system for dengue in Columbia.
Minor improvement in english is suggested for better flow of the MS.
Author Response
The study concludes that monitoring liver enzymes, neutrophils, platelets, leukocytes, etc. improves the prediction of severe dengue risk. However, it does not address other critical risk factors, such as viral serotypes, viral load, and host immune responses, which could enhance prediction accuracy. We acknowledge that factors such as viral serotype, viral load, and host immune response play a crucial role in dengue severity. However, our study focused on routinely available clinical and laboratory parameters to enhance the practicality of early risk prediction in real-world settings. While incorporating virological and immunological markers could improve prediction accuracy, such data was not available for all patients in our study. We have now acknowledged this limitation in the discussion section and highlighted the need for future studies integrating these additional factors. Study Sites and Sampling: The methods section needs further clarification on the rationale for site selection. It would be beneficial if the chosen sites were representative of Colombia as a whole. Additionally, the selection criteria for prospective and retrospective samples should be explicitly stated. While we acknowledge that broader geographic representation could enhance generalizability, logistical constraints and data availability limited our study to this region.Regarding sampling, we have now explicitly stated the selection criteria for both prospective and retrospective samples in the methods section. The retrospective data were obtained from existing medical records, while the prospective cohort included patients meeting predefined inclusion criteria during the study period. These clarifications have been clarified into the manuscript.
Background: Briefly mentioning the clinical burden of severe dengue in Colombia would strengthen the study’s justification. We agree that clarifying the rationale for site selection and sampling criteria is important. This information is already detailed in the manuscript (lines 76–79). However, we will ensure that it is explicitly stated and easily accessible within the introduction to enhance clarity. Machine Learning Approach: Given that the authors acknowledge issues with retrospective data quality, is a machine learning approach appropriate for this setting? Would a hybrid model integrating traditional epidemiological methods with ML offer a better balance between accuracy and interpretability? Thank you for your suggestion. While machine learning could enhance prediction, data quality limitations made it unfeasible for this study. Our approach focused on practical, real-world applicability using available clinical data. In the discussion section we acknowledge its potential for future research with higher-quality prospective data. Geographic Representation: Including maps of Colombia showing study sites would improve clarity and contextual understanding. Due to copyright restrictions on shapefiles, we are unable to include maps. However, we have provided the geographic coordinates of the study sites to enhance clarity and contextual understanding. Clinical Implications: The clinical significance of the findings should be emphasized. How do they enhance current triage protocols? Our findings have significant clinical implications, particularly in low-complexity and rural healthcare settings where access to advanced diagnostic tools is limited. By identifying key predictors of severe dengue (SD) using routine laboratory tests, our model (AUC = 0.91) provides a practical tool for early risk assessment and triage in resource-limited environments. This can help optimize patient referrals, improve resource allocation, and support timely interventions, especially during outbreaks. We have strengthened this discussion in the manuscript. Model Application in Clinical Settings: Instead of broadly stating that the model is useful for triage, specify how it could be integrated into clinical workflows. The discussion in lines 463-474 already addresses aspects of how the model can be integrated into clinical workflows. However, we have refined this section to explicitly highlight its role in triage, early risk assessment, and referral optimization in low-complexity and rural settings for greater clarity. Secondary Infection and Antibody-Dependent Enhancement (ADE): The study notes that secondary infections are associated with severe dengue. Could the authors elaborate on whether this is linked to ADE? For reference, DENV-3 in Nicaragua has been associated with more severe primary infections, as well as in India and Mexico (Lancet, October 25, 2024).The discussion in lines 435-446 addressed this topic, but we have refined this section for clarity.
Our study confirms the link between secondary infections and severe dengue, potentially related to antibody-dependent enhancement (ADE), though we lacked immunological markers to assess it directly. The association of DENV-3 with severe primary infections in Nicaragua, India, and Mexico (Lancet, October 25, 2024) underscores the need for serotype-specific analyses. However, limited serotype data prevented further exploration. We have strengthened this discussion in the manuscript.Figure 2: The figure is difficult to interpret. Consider providing a clearer explanation in the legend or removing it if it does not add substantial value.
Thank you for your comment. We acknowledge that Multiple Correspondence Analysis (MCA) can be challenging to interpret, but it provides valuable insights for the study. Its interpretation is already detailed in the Results section, where we explain its relevance and contribution.The discussion section could be more concise. Some parts are overly detailed and could be streamlined for better readability.
We have carefully reviewed the discussion to ensure clarity and relevance while maintaining the necessary details to support our findings. We believe the current level of detail is essential for a comprehensive understanding of the study’s implications.The current early warning system for dengue in Colombia.
Colombia's Early Warning and Response System (EWARS) leverages climatic and epidemiological data to predict dengue outbreaks, allowing timely interventions. While effective, challenges remain in real-time data quality and integration of socio-economic factors. We have added a brief mention of this system in the discussion for context.
Reviewer 4 Report
Comments and Suggestions for Authors
I really liked the article that deals with qualitative and quantitative variables to predict the risk of dengue in a hyperendemic region of Colombia. The authors emphasize the needs to seek comprehensive validation for the use of predictive models aimed at assessing risck factors primarily associated with severe dengue. And is this way, they aimed to develop a predictive model for severe dengue using data from Urubá , a hyperendemic region of Colombia.
The use of the logistic regression model for qualitative variables dichotomous ones such as yes and no, positive and negative, or nominal ones like sex, among others as well as the odds ratio resulting from the aforementioned analysis is not a novelty nor are other statistical treatments. However, the authors demonstrated a strong understanding of the inferences used for both qualitative variabels and continous and discrete quantitative variables. the authors were very thorough in recording the eligibility and exclusion criteria for the variables. The entire methodology was very well detailed.
All references have been used and are properly distributed throughout the body of the article. References 56 and 57 are located at the end of page 16, in the Institutional Review Board Statement Section. Could they be placed together with the mentioned section in Materials and Methods?
The results were clearly presented, distributed in tables and two figures. A graphical representation showing the relationships betweeen the data categories and the ROC curve of logistic regression.
The discussion presents the interpretation of the main results and their relation to similiar studies. An evaluation of the research was also made, highlighting the study's limitations.
And finally, considering that the implementatiton of predictive models for the risk of severe dengue in real clinical settings, especially in Colombia is limited. I Believe the authors are indeed making a valuable contribution to local health services in Colombia and to peers working with predictive models for dengue risk.
Author Response
All references have been used and are properly distributed throughout the body of the article. References 56 and 57 are located at the end of page 16, in the Institutional Review Board Statement Section. Could they be placed together with the mentioned section in Materials and Methods? Some lines about the ethical aspect, along with the mentioned references, are included in the materials and methods section. However, they are also retained in the institutional review board statement section to support this journal's requirement.Reviewer 5 Report
Comments and Suggestions for Authors
This is an interesting and comprehensive study based on a well-designed analysis of many cases of Colombian patients infected with DENV. I like it a lot because this study provides a valuable tool to detect severe cases of dengue in humans prior to symptoms of high impact on their health.
I believe this model will allow the healthcare provider to easily and quickly assess and prioritize serious dengue risks, especially during outbreaks. I thank a new way to control the growing number of deaths from severe dengue infection in Latin America.
I have also some minor suggestions:
-Please have an English language expert polish your writing.
-Remember to write each scientific name in italics throughout the text (pay special attention to References section).
-Reference No. 6 is missing the year of publication.
Comments on the Quality of English LanguagePlease have an English language expert polish your writing.
Author Response
Please have an English language expert polish your writing. We have carefully reviewed and refined the language throughout the manuscript to enhance clarity and readability. Additionally, we have consulted a native English speaker to ensure the writing meets high linguistic and academic standards. Remember to write each scientific name in italics throughout the text (pay special attention to References section). We have carefully reviewed the manuscript and ensured that all scientific names are properly italicized throughout the text, including the References section. Reference No. 6 is missing the year of publication. Thank you for your observation. We have corrected Reference No. 6 by including the year of publication to ensure accuracy and completeness.Reviewer 6 Report
Comments and Suggestions for Authors/
Author Response
Done
Round 2
Reviewer 3 Report
Comments and Suggestions for Authors
Thank you for thoroughly addressing my suggestions. I have no further comments.
Reviewer 6 Report
Comments and Suggestions for Authors
This manuscript"Dengue severity prediction in a hyperendemic region in Colombia" has certain reference value for clinical management of dengue cases,Despite the use of multiple machine learning algorithms (such as logistic regression, decision trees, and random forests) and a model with an AUC value of 0.91 that performs well, the study makes no mention of external validation in different populations or different regions.Some of the study's data collection overlapped with the COVID-19 outbreak, which may have led to underreporting or misdiagnosis of dengue cases, which may have influenced the study's results. The epidemic interference affected the representativeness of the data.The author has discussed some of the above issues, considering that this study was conducted in Colombia, Latin America, although there are some shortcomings, it is recommended to be published.
Comments on the Quality of English Language/